# A novel approach to interrogating the effects of chemical warfare agent exposure using organ-on-a-chip technology and multiomic analysis

Tyler D. P. Goralski[1]ʘ*, Conor C. Jenkins[1]ʘ, Daniel J. Angelini[1], Jennifer R. Horsmon[1], Elizabeth S. Dhummakupt[1], Gabrielle M. Rizzo[1], Brooke L. Simmons[2], Alvin T. Liem[3], Pierce A. Roth[3], Mark A. Karavis[1], Jessica M. Hill[3], Jennifer W. Sekowski[1], Kyle P. Glover[1]

1 U.S. Army Combat Capabilities Development Command (DEVCOM) Chemical Biological Center (CBC), Gunpowder, MD, United States of America, 2 Oak Ridge Institute for Science and Education (ORISE), Oak Ridge, TN, United States of America, 3 DCS Corporation, Belcamp, MD, United States of America

ʘ These authors contributed equally to this work.
* tyler.d.goralski.civ@army.mil

**Data Availability Statement:** The mass spectrometry metabolomics data have been deposited to the Metabolights repository (https://

## Abstract

Organ-on-a-chip platforms are utilized in global bioanalytical and toxicological studies as a way to reduce materials and increase throughput as compared to *in vivo* based experiments. These platforms bridge the infrastructure and regulatory gaps between *in vivo* animal work and human systems, with models that exemplify active biological pathways. In conjunction with the advent of increased capabilities associated with next generation sequencing and mass spectrometry based '-omic' technologies, organ-on-a-chip platforms provide an excellent opportunity to investigate the global changes at multiple biological levels, including the transcriptome, proteome and metabolome. When investigated concurrently, a complete profile of cellular and regulatory perturbations can be characterized following treatment with specific agonists. In this study, global effects were observed and analyzed following liver chip exposure to the chemical warfare agent, VX. Even though the primary mechanism of action of VX (i.e. acetylcholinesterase inhibition) is well characterized, recent *in vivo* studies suggest additional protein binding partners that are implicated in metabolism and cellular energetic pathways. In addition, secondary toxicity associated with peripheral organ systems, especially in human tissues, is not well defined. Our results demonstrate the potential of utilizing an organ-on-a-chip platform as a surrogate system to traditional *in vivo* studies. This is realized by specifically indicating significant dysregulation of several cellular processes in response to VX exposure including but not limited to amino acid synthesis, drug metabolism, and energetics pathways.

## Introduction

Traditionally, researchers have relied heavily on animal models and two-dimensional tissue cultures for toxicological predictions [1,2]; however, *in vivo* work is time consuming and

www.ebi.ac.uk/metabolights/) with the dataset identifier MTBLS5466. The mass spectrometry proteomics data have been deposited to the ProteomeXchange Consortium (http://proteomecentral.proteomexchange.org) via the PRIDE database with the dataset identifier PXD037722. The RNA sequencing transcriptomics data have been deposited to the NCBI Gene Expression Omnibus (GEO) repository (https://www.ncbi.nlm.nih.gov/geo/), with the dataset identifier GSE221207.

**Funding:** KG CB10735 Defense Threat Reduction Agency https://www.dtra.mil/ The funders played a role in the decision to publish. KG BQ5066 Defense Advanced Research Projects Agency https://www.darpa.mil/ The funders played a role in the decision to publish.

**Competing interests:** The authors have declared that no competing interests exist.

incurs a high financial burden to obtain statistically valid results. More importantly, human hepatic tissue cultures grown on dishes or plates often lack physiological accuracy and can vary from donor to donor [3]. For these reasons, microphysiological systems (MPSs) have been gaining traction as more accurate and potentially more cost effective means for predicting human outcomes in response to drug exposures or other compounds of interest [4,5]. By providing a three dimensional architecture comprised of a multi-cell type microenvironment complete with fluid flow and other cellular and environmental dynamics present in a functional organ, MPSs are leading the charge for alternative *in vivo* toxicity testing [6].

Recently, organ-on-a-chip technologies have successfully recapitulated the physiological as well as the phenotypic characterisitcs of a number of different organs (e.g. brain, liver, lung, intestine, and kidney) [4,6–11]. Each of these unique systems relies on specialized components and features that allow for organ mimicry surpassing traditional tissue culture. With respect to the Emulate system used in this study, tissue chips are comprised of both a top and bottom channel seperated by a porous membrane, which house the different cell types and microvasculature associated with the amenable organ types [5]. The liver chip is designed to maintain hepatic-specific cytoarchitecture as well as function. The tissue chips are also subjected to constant, unilateral flow. This can be useful for facilitating exposures from exogenous compounds as well as subjecting the cells with sheer stresses comparable to that experienced *in vivo*. Four different cell types of hepatic origin are maintained in this model They consist of hepatocytes in the top channel and a layer composed of the nonparenchymal (NPC) Kupffer cells, human stellate cells (HSCs), and liver sinusoidal endothelial cells (LSECs) in the bottom channel [5]. The chips are housed in Pods, which function to supply medium to the various cell types. Cells are dosed through the inlet resevoirs of the Pod, and effluent is collected from the outlet resevoirs for analysis. The flow rate of the medium through the top and bottom channels of the chip is controlled by the Zoë. The Zoë accomodates a pump manifold, which is engaged with the chips housed in Pods to produce the medium flow. All of the gas exchange within the chips is controlled by a device called an Orb [12].

RNA sequencing and mass spectrometry-based '-omic' technologies have experienced a renaissance in the past decade. Small mass sample extraction methods, instrument resolution and mass accuracy have been increasing, allowing for a deeper and more comprehensive understanding of the biological systems impacted by a perturbation [13]. Furthermore, growth in high-resolution mass spectrometry (HRMS) has enabled new insights for characterization, peptide sequence determination, post-translational modifications, and small molecule profiling through untargeted proteomics and metabolomics acquisitions. The physiological accuracy of organ-on-a-chip systems complements the capabilities of HRMS and provides an excellent platform for transcriptomic, proteomic and metabolomics analyses. There are, however, limitations to the various organ-on-a-chip platforms, such as total cell density. The total cell number per Emulate liver chip is considerably lower than traditional tissue culture, which was overcome in this study through a sensitive means of extracting transcripts, proteins, and metabolites from lysates with what would normally be considered insufficient yield. While others have analyzed the host transcriptome and proteome of organ chip tissues before [12], we sought to provide a simple and reproducible protocol for extracting the transcripts, proteins, and metabolites from small sample sizes, including digestion of proteins for bottom-up analysis.

Using this protocol, we sought to explore aspects of VX toxicity that have previously been difficult to capture using either *in vivo* models or rudimentaty cell culture systems. VX is a highly potent organophosphate nerve agent that causes acute lethal toxic effects via inhibition of acetylcholinesterase (AchE); however, there are a number of studies that provide evidence for additional molecular targets that may contribute to VX toxicity [14–17]. For example, we

recently discovered through proteomic and metabolomic analyses that VX binds 132 different proteins identified from hairless guinnea pig serum, of which were a multitude of enzymes involved in metabolism and cellular energetic pathways [18]. We therefore sought to utilize a newly developed protocol for assessing multiomic samples from human tissue chips to both validate these findings while also assessing for other organ specific perturbations that may be occuring in the liver following VX exposure. Untargeted metabolomics showed similar upregulation of choline and phosphocholine which reflect current *in vivo* findings. Untargeted proteomics and transcriptomics show an enrichment of pathways associated with drug metabolism, as well as perturbations to energetics pathways, demonstrating the chip system is properly modeling the liver processes post exposure to the chemical warfare agent, in addition to pathways akin to the method of action of VX that has been previously observed. Taken together, these data suggest that the novel approach of combining organ-on-a-chip technology with global multi-omics anaylsis is a viable means for assessing the effects of compounds or agents of interest on human organ tissues.

## Results

### Chip tissue viability and function

In order to determine an acceptable, sub-lethal dose range for VX in liver chips that would provide protein, gene transcript, and metabolite dysregulation without cell death, HepG2 liver cell cultures were subjected to two-fold increasing concentrations of VX for 24 hours, and an LDH cytotoxicity assay was performed (Fig 1A). At 62.5 μM, cytotoxicity of greater than 20% was observed. No significant cytotoxicity in liver chips was observed at lower concentrations compared to the negative control (-C). Based on these data, 25 μM, 12.5 μM, and 6.25 μM were selected for the three VX exposure concentrations in the liver chip study to ensure that the highest concentration was well below the cytotoxic range. An albumin ELISA performed after the liver chip VX exposures indicated normally functioning hepatocytes, as the treated chip albumin output equally matched that of the untreated controls (Fig 1B). An LDH cytotoxicity assay was conducted on effluent collected from the VX exposed liver chips, however, no cytotoxicity was observed (S1 Fig). Brightfield images taken throughout the VX exposures of both the hepatocytes and NPCs indicated a healthy monolayer with no cell debris (Fig 1C and 1F). Chips treated with the highest concentration of VX were fixed and stained. Hepatocyte MitoTraker staining for mitochondria (Fig 1D) and LSEC phalloidin staining for actin (Fig 1E) demonstrate viable liver tissues and liver microvasculature.

### Metabolomics

Post-acquisition analysis of the untargeted metabolomics of the liver chips showed 974 metabolites identified by reverse phase (RP) positive acquisition, 328 by RP negative, 601 by Hydrophilic interaction chromatography (HILIC) positive, and 115 by HILIC negative. The heat map in Fig 2 displays the metabolomic expression values of the RP positive acquisition. Samples taken Day 1 post-exposure and Day 7 post exposure group independently based on their metabolic profile. This observation was consistent for the additional acquisitions.

Sparse Partial-Least Square Discriminant Analysis (sPLS-DA) is a tool that has shown great fidelity in the feature selection process pertaining to the features that can discriminate between multiple classes [19]. sPLS-DA was performed in Compound Discoverer utilizing two components to determine the top ten compounds that are contributing most to the differentiation based on exposure concentration, independent of day post exposure (Fig 3). Of these compounds, pentahomomethionine, 1,4-dideoxyhexitol and choline were significantly (p-val < = 0.05) dysregulated after VX exposure compared to the untreated liver chips. Choline is of high

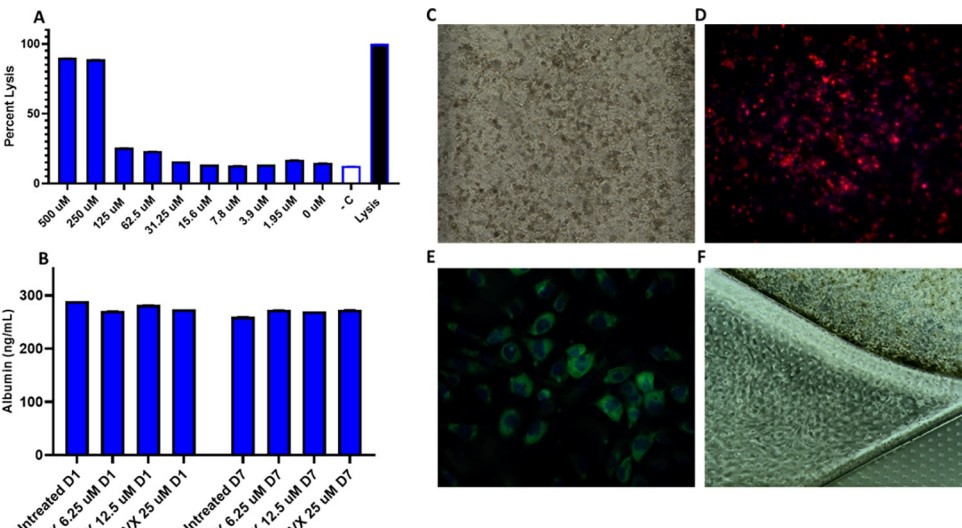

**Fig 1.** A. LDH cytotoxicity assay showing the effects of VX on HepG2 liver cells after 24 hours. B. Albumin ELISA showing no effect on hepatic function. C. Brightfield image of hepatocyte layer. D. MitoTraker stain of hepatocytes showing viability. E. Phalloidin stain showing F-Actin in LSECs. F. Brightfield image of the interface between the top and bottom channels of the chip.

interest as it has been shown in previous studies to be involved in the method of action of VX. Choline is upregulated with VX exposure (Fig 4) which has also been observed in vivo [18]. Both compounds 1,4-dideoxyhexitol and pentahomomethionine are involved in glucosinolate biosynthesis and 2-oxocarboxylic acid metabolism; both of these pathways are associated with the tricarboxylic acid (TCA) cycle, which has been shown to be impacted by VX exposure. When performing the sPLS-DA analysis on only Day 1 samples, pentahomomethionine was conserved, which reflects its importance in discriminating from baseline at Day 1. A compound similar to maltotriose was also observed via sPLS-DA, demonstrating perturbation in the small molecule sugars of the system. Previously published studies of human exposed to organophosphate-based pesticides indicate these chemicals as potentially contributing to increased incidence of disrupted glucose metabolism and diabetes. A quinolone carboxylic acid was also identified, which has been shown to impact cytochrome p450. Data acquired by HILIC positive mode and analyzed by sPLS-DA revealed phosphocholine, an intermediate of phosphatidyl choline, as a compound contributing highly to the differentiation of exposure concentrations, independent of day. Part of the MOA of VX toxicity is the irreversible inactivation of the acetyl cholinesterase (AChE) junction, which causes toxic levels of acetylcholine (Ach) to accumulate. Choline, depending on pathway pressures, can either be converted to phosphocholine or to acetylcholine. Comparing the boxplot of the phosphocholine (Fig 4B) to the boxplot of choline (Fig 4A), there appears to be an inhibition of the conversion of choline to phosphocholine, which aligns with the MOA of VX as the pathway works to overcome the inactivation of the AchE junction. Erucamide was also identified, which is involved in ammonia recycling. The ammonia recycling pathway reroutes ammonia from urine and recycles it within the body for nitrogen metabolism, and this pathway has been shown to be disrupted due to exposure of organophosphates.

When a secondary sPLS-DA was performed with additional discrimination by hours post exposure, the discrimination introduced an observation of plasticizer contamination (S2 Fig). Octyl isocyanate is a compound utilized to improve the tensile strength of poly vinyl alcohol and helps control water vapor permeation. Monoisocyanates appear to exert toxic effects,

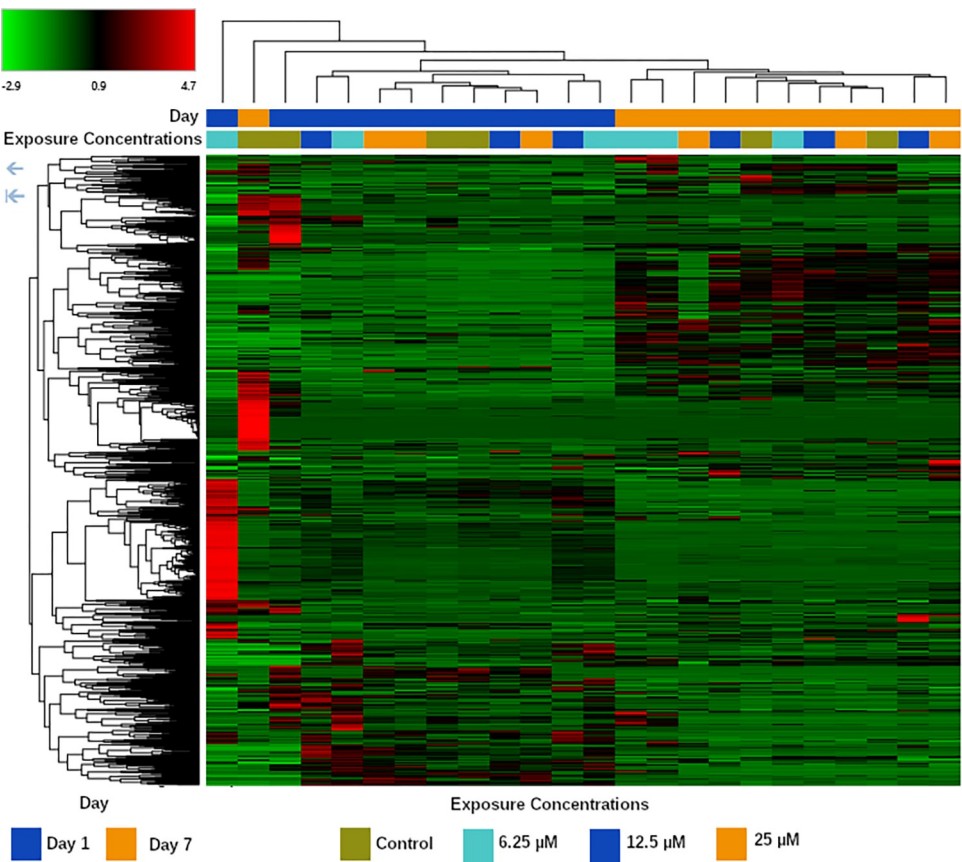

**Fig 2. Heat map of the feature abundances found in the reverse phase positive mode acquisition.** Clustering was performed based on the euclidean distance of the median values of each feature. Color Scaling was based on the Log2 Fold Change over untreated samples.

including delayed lethality, which is similar to those induced by methyl isocyanate. This is extremely toxic to humans. Styrene, a possible carcinogen as declared by the World Health Organization (WHO), also appeared with the sPLS-DA analysis, which is used predominately in the production of plastics and resins. DIPEA (Hunig's base) was shown as a top compound with discrimination abilities pertaining to solely day discrimination. This is a good base/poor nucleophile that is used to deprotonate esters which is integral in plastic, resin and lacquer development and also a cancer and reproductive hazard as well as lung irritant.

## Proteomics

The proteomics abundance values cluster well amongst each of the bioreplicates and the day post exposure (Fig 5). The controls from both days group close together indicating that batch effects of letting the chips grow for an extended period have less effect on the proteome of each sampling group.

Recent literature regarding mechanism of action of VX indicated direct impacts on the TCA cycle [18]. Performing a network analysis detailed an interconnection of the significantly changing (pval = 0.005) proteins, centered on glyceraldehyde-3-phosphate dehydrogenase (Fig 6), an integral member in the glycolysis pathway [20]. Also within this network, additional proteins involved in the TCA cycle were identified. For example, GDP-forming succinate-CoA ligase experienced a significant downregulation 1d post VX exposure. Malate

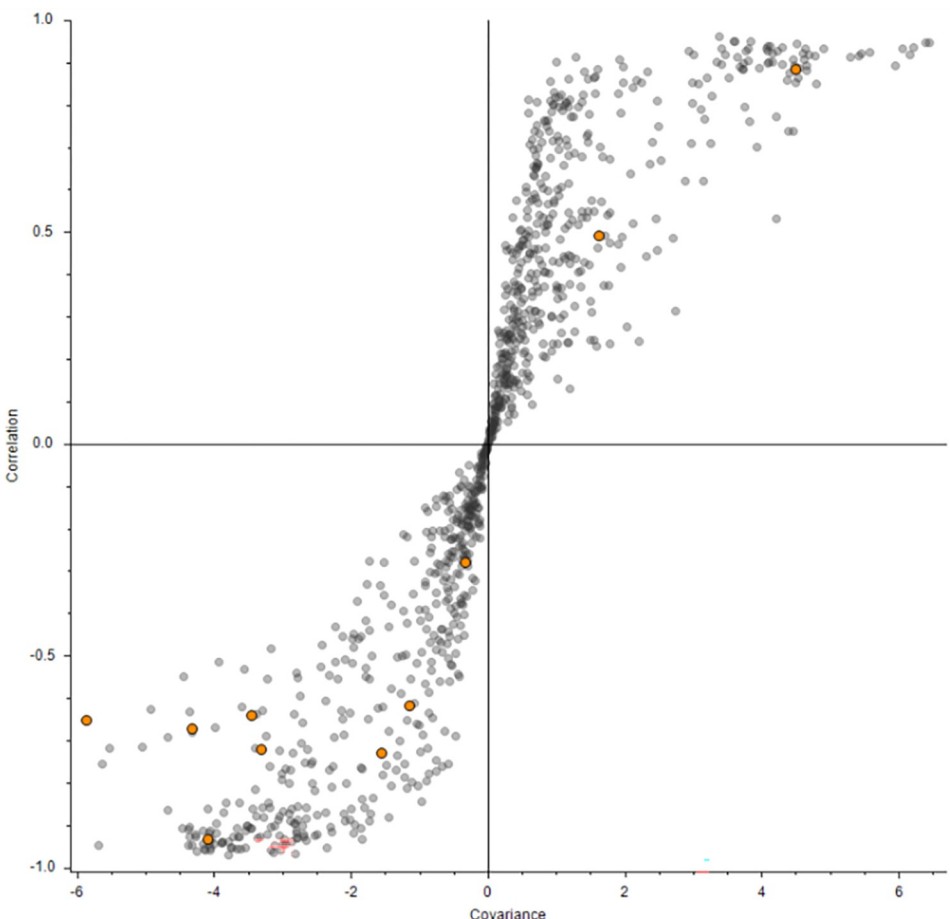

**Fig 3. sPLS-DA of the reverse phase positive metabolomics data.** The second component of the sPLS-DA Analysis details the ten compounds contributing most to the classification of the Exposure concentrations.

dehydrogenase also experienced a dysregulation; it is initially significantly downregulated 1d following exposure, but then appears to recover and become upregulated by 7d post-exposure. Additionally, 3-ketoacyl-CoA thiolase was shown to be significantly downregulated 1d following VX exposure. This protein is involved in beta-oxidation, a process that breaks down fatty acids for entry into the TCA cycle.

## Transcriptomics

As displayed in the analysis of the proteomics and metabolomics data, the transcriptomic data also clustered well each day post exposure as well as the bioreplicate set for each exposure concentration (Fig 7). It is worth noting that the non-exposed controls from both timepoints also grouped closely. This indicated that there was little variation between different chip preparation lots and that the effects observed in the VX exposed liver chips was due to VX.

While low p-values for these key gene expression changes using linear regression analyses were observed, due to the low biological replicate number in this study, the q-values (or false discovery rate, FDR) were higher than acceptable for making clear mechanistic inferences using gene expression alone. In order to overcome this issue, these data was analyzed using Gene Set Enrichment Analysis (GSEA). This analysis is a statistical tool that compares an

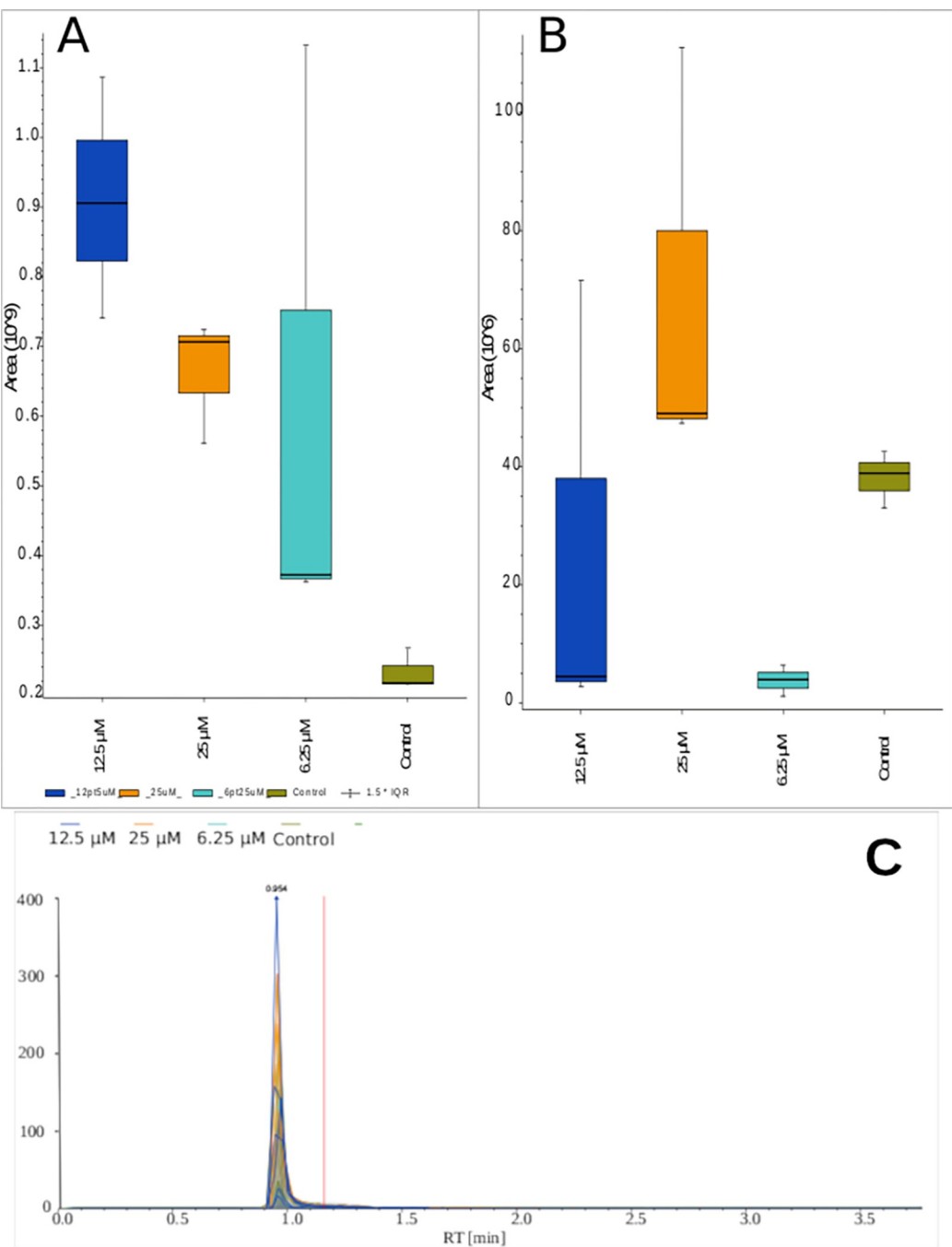

**Fig 4.** Boxplot of choline expression values based on exposure independent of day (A). Chromatogram of choline in the RP positive acquisition (B). Phosphocholine Boxplot (C).

experimentally derived gene set with curated and annotated gene sets from databases such as MsigDB, Reactome, and GO, and assigns a statistical score to the relevance of each curated pathway to the experimental gene expression set [21–23]. When the transcriptomic data were analyzed using GSEA and compared with the metabolomics and proteomics data analyses, a number of statistically sound biological pathways were revealed [24].

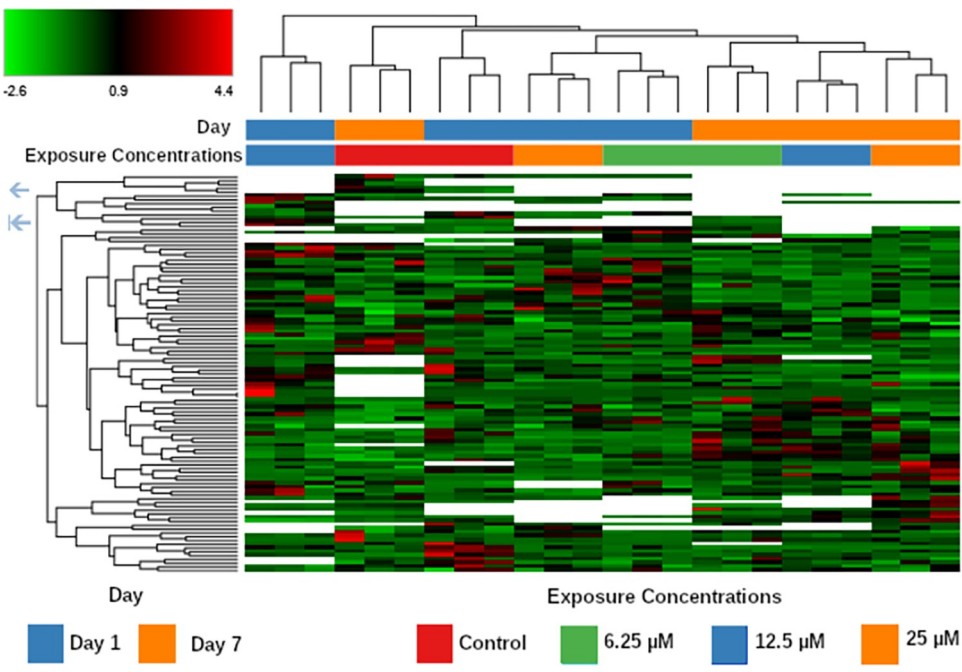

**Fig 5. Cluster map of the proteomic expression values showing clustering of the control treatments (red) and the 6.25 exposure level (green).** Higher exposure levels show poor clustering indicative of potential cell death events leading to high variation. Sample clustering by day also shows good differentiation. Color Scaling was based on the Log2 Fold Change over untreated samples.

Based on the linear regression analyses, the GSEA was performed on the Day 1 and Day 7 transcriptomics data sets separately. GSEA carried out on the Day 1 set revealed 11 pathways (Table 1) with a p-value of $\leq 0.1$ and reasonable NES scores. All the genes shown below in the heat map are statistically significant with a p-value $< 0.05$ during linear regression and overlap with our gene set (Fig 8).

Not surprisingly, many of the significant pathways revealed by the transcriptomic data are functionally related to those observed from the proteomic and metabolomic data. As shown in the Pathcards pathway network (Fig 9), the TCA pathway shares functional links with other energy-metabolism related pathways including the Respiratory Electron Transport, Electron Transport Chain (ETC), and Oxidative Phosphorylation pathways. In the ETC, mitochondrial function directly participate in the regulation of Reactive Oxygen Species (ROS) [25]. Our data show that the upregulation of the antioxidant response protein, catalase (CAT gene), at Day 1 suggests a cellular reaction to ROS (i.e. $H_2O_2$ production) in the exposed cells [26]. Other genes such as NFKBIA, IFNGR2 and TLR2 also support the cells' response to the generation of ROS and oxidative stress. Genes upregulated within the GSEA Mitochondrial pathways have other known functions such as reacting to ROS (XIAP), promoting apoptosis (APAF1, CASP8, AIFM1), and countering apoptotic pathways (XIAP, BCL2, BIRC2 and BIRC3). Also countering ROS, Glutathione peroxidase 7 protein (GPx7) appears to be reacting to detoxify hydrogen peroxide present in the cells [27]. The increase in ROS in cells is also related to mitochondrial dysfunction [28].

## Discussion and conclusion

Collectively, this work presents and develops methodology to extract low yield samples from an organ-on-a-chip model and process them for untargeted proteomic, metabolomic, and

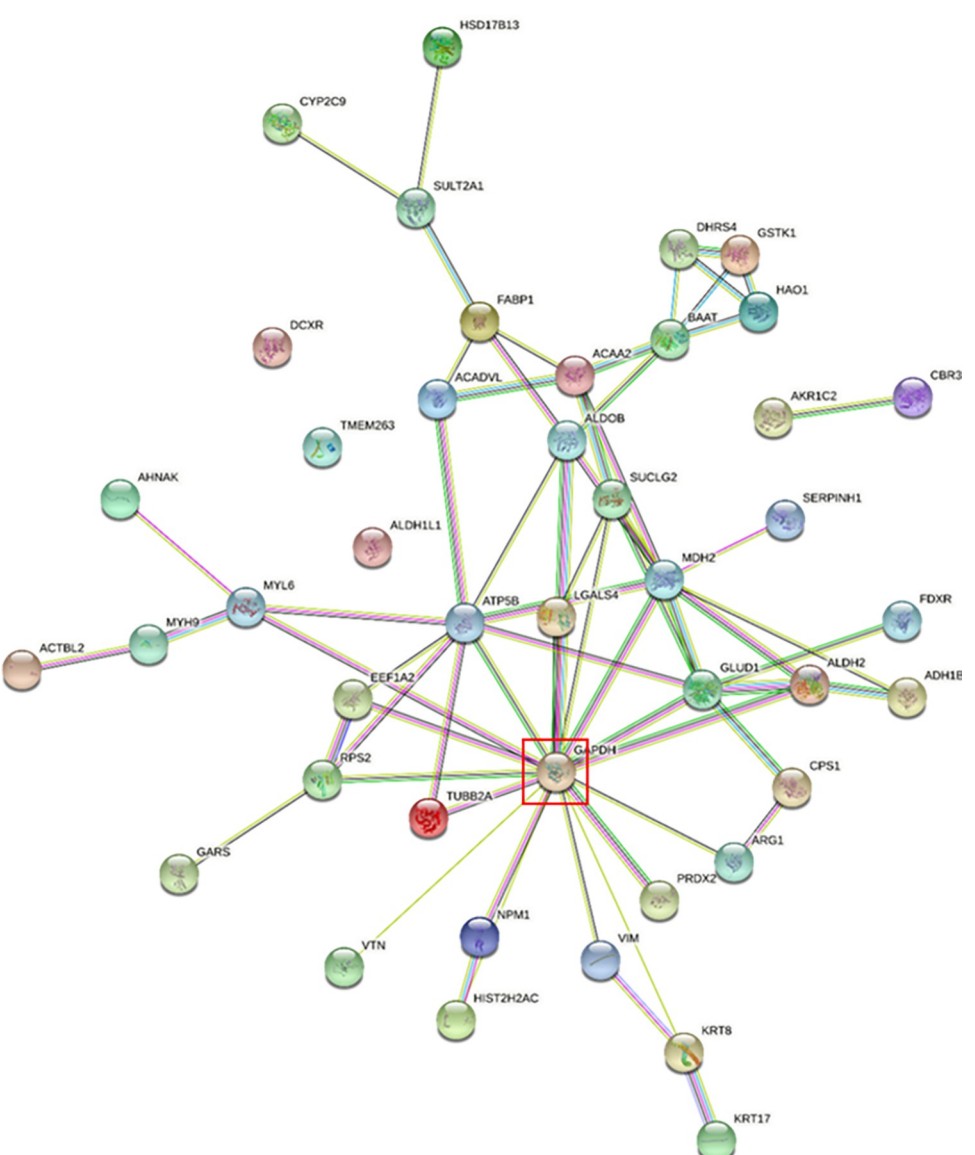

**Fig 6. Network analysis of significantly changing proteins.** GADPH is highlighted by a red box.

transcriptomic analysis. Despite the relatively low abundance of material for analysis (~100,000 cells per chip) compared to conventional cell culture methods, solid coverage of the proteome, metabolome, and transcriptome were obtained in the current samples. Utilizing label free quantitative methodology at both the proteomic and metabolomic levels as well as total RNA-Seq analysis, we verified that the biological effects of VX exposure persist with utilization of the human liver chip system.

Through advancement of this novel methodology, we were successful at identifying proteins, metabolites, and expressed genes supporting pathways previously implicated in VX exposure *in vivo* [18] and obtained unique findings on the effects of VX on liver tissues. One example of a previously identified metabolite, choline, was upregulated in VX exposed liver chips and is implicated in a potential secondary MOA. A number of different compounds associated with the TCA cycle and glycolysis were also disrupted in both studies, suggesting

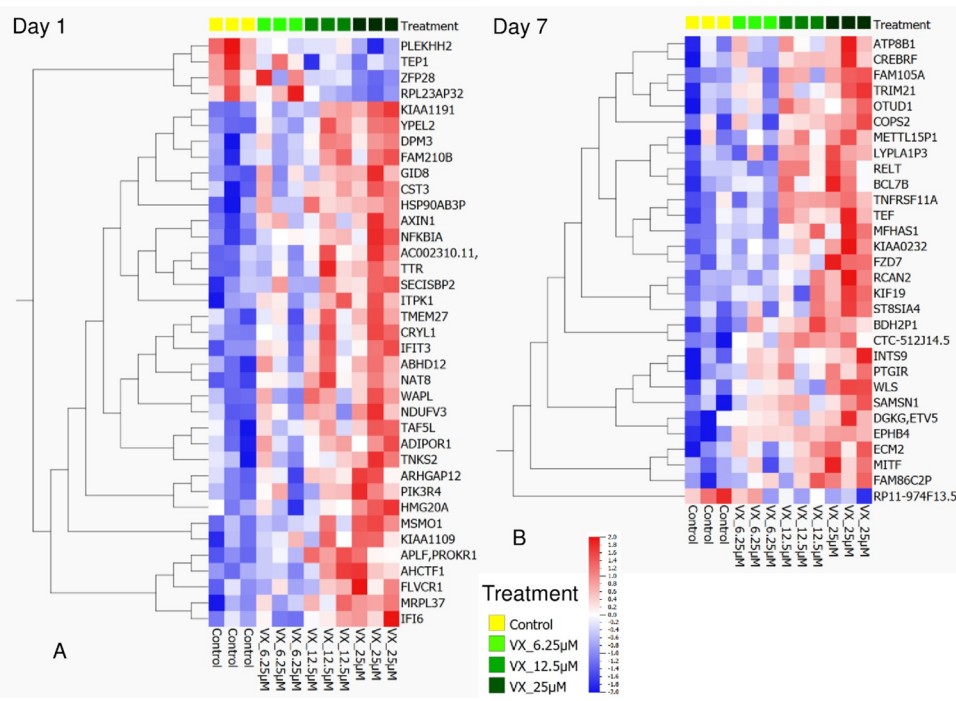

**Fig 7. Transcriptomics heat map from linear regression analysis.** A) Day 1 only changes in expression, described by 37 genes (p = 0.005). B) Day 7 only changes in expression, described by 30 genes (p = 0.005). Linear regression analyses revealed that the transcriptomics data are more clearly described by examining Day 1 and Day 7 gene expression changes separately.

that exposure to VX effects the tissue's energy acquisition pathways. We found additional related evidence of perturbations in the regulation of ROS, which is also linked to downstream mitochondrial energy metabolism functions [28]. It is interesting to note that we identified dysregulation in the liver cells' abilities to recycle toxic waste products such as ammonia. This could be compounded due to potential dysregulation of cytochrome p450 function, as evidenced by a reduction in a quinolone carboxylic acid associated with the enzyme. Amino acid production and degradation as well as tRNA biosynthesis was also dysregulated; these alterations indicate a potential effect on overall mitochondrial performance. These results also suggest that some cell signaling could also be down regulated due to exposure to VX.

**Table 1. GSEA pathways most significant for the Day 1 transcriptomics data.**

| PATHWAY | P-value | Q-value | NES |
|---|---|---|---|
| KEGG_GLUTATHIONE_METABOLISM | 0.006 | .330 | 1.69 |
| HALLMARK REACTIVE OXYGEN SPECIES | 0.018 | .386 | 1.55 |
| GOBP_CELLULAR RESPONSE TO AMMONIUM ION | 0.025 | .514 | 1.38 |
| BIOCARTA_MITOCHONDRIA_PATHWAY | 0.026 | .380 | 1.69 |
| HALLMARK FATTY ACID METABOLISM | 0.051 | .565 | 1.59 |
| BIOCARTA_ETC_PATHWAY | 0.064 | .511 | 1.41 |
| DNA-RNA_METABOLISM | 0.071 | .379 | 1.59 |
| HALLMARK OXIDATIVE PHOSPHORYLATION | 0.076 | .452 | 1.51 |
| HALLMARK INFLAMMATORY RESPONSE | .0990 | .510 | 1.4 |
| HALLMARK GLYCOLYSIS | 0.103 | .468 | 1.33 |
| NITROGEN COMPOUND METABOLIC PROCESS | 0.105 | .387 | 1.48 |

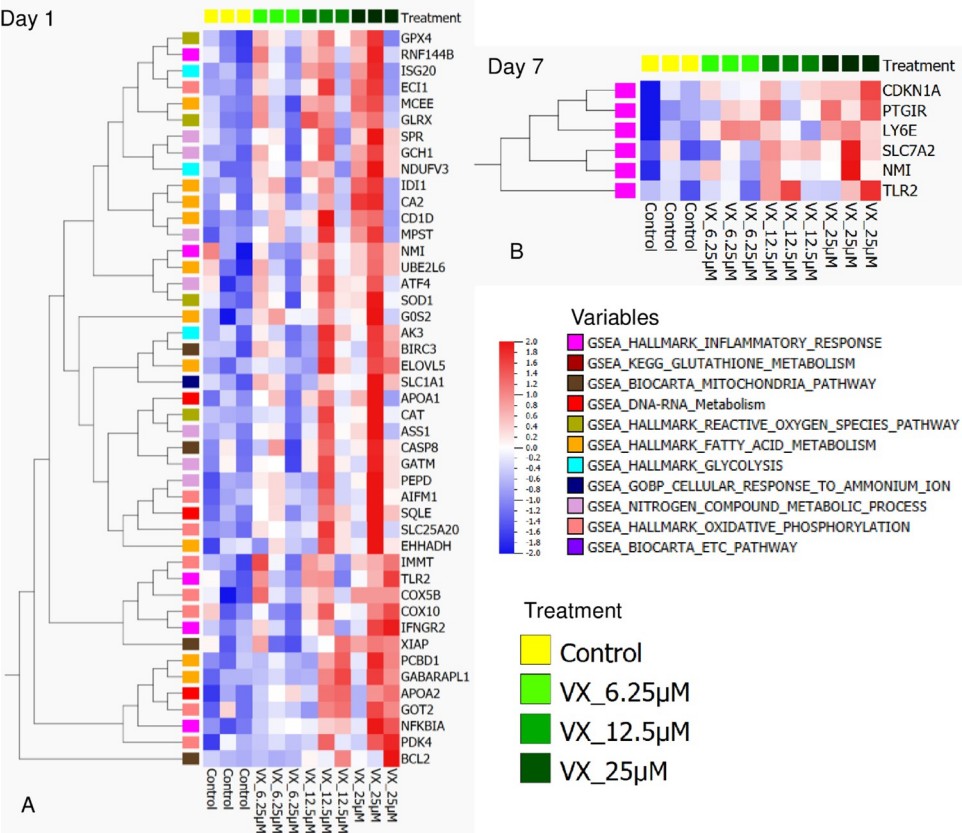

**Fig 8. Transcriptomics heat map of GSEA pathways on Days -1 and 7-post exposure by (P = 0.005, *q = 0.87).** *high q-value is due to low bioreplicate numbers.

Despite the success of the chip exposure and data acquisition pipeline described in this study, we still found a few challenges concerning chip-to-chip and day-to-day variability. While the Day 1 and Day 7 data grouped well within their respective replicates for both VX

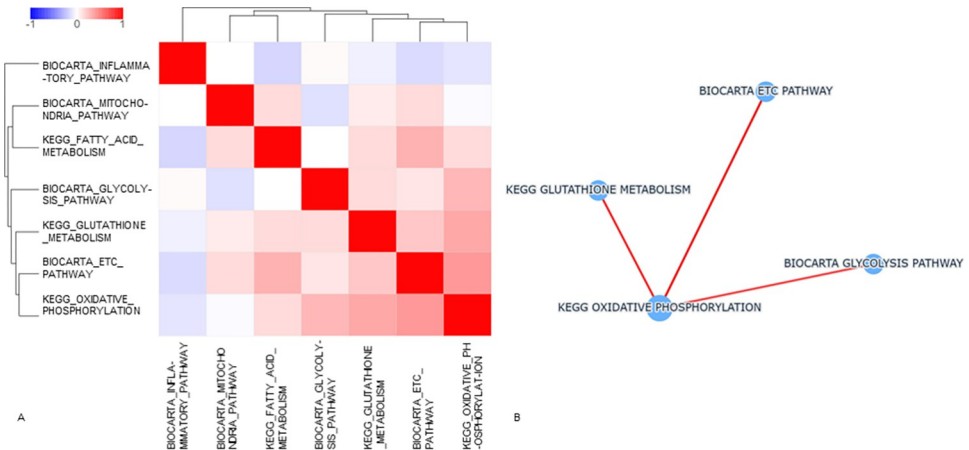

**Fig 9. PathCards pathway network illustrating linkages between pathways identified by proteomic, metabolomic and transcriptomic data in our work.**

exposed and untreated chips, there were significant metabolic differences observed between the untreated Day 1 and untreated Day 7 chips. One possible explanation is that basal level activity within the tissues could mature the longer they are in the chips. It is also possible that the inconsistency between time points could be due to slight differences in total cell number present in each sample. This is also exacerbated between the untreated chips, which would make it difficult to normalize the data sets to our control samples. In future organ-on-a-chip studies, normalization must be performed to decrease chip variability between days of exposure. Additionally, the possibility exists that the presence of plasticizers and other plastic and resin production chemicals, could be present in the chips and induce off target effects on the system that can lead to incorrect assumptions of VX exposure and MOA. Given that the plasticizers are a likely byproduct of the organ chip systems, it is improbable that these off target effects would be observed in multiomics data sets obtained from similar exposures *in vivo*. However, in our assessment, effects that would implicate the specific plasticizers identified by omics analysis were not reflected in our data.

In total, this work evaluates the combination of organ-on-a-chip technology with global multi-omics analysis and demonstrates its potential usefulness in performing toxicological assessments. These results leave open the possibility for expansion of this methodology towards studying additional organ systems and compounds of interest. It is promising that the data generated align with previous published *in vivo* multi-omics studies, which demonstrates this method as a potential viable means for collecting physiologically accurate data that complements existing *in vivo* model data for toxicological assessments.

## Materials and methods

### Reagent preparation

Human hepatocytes, HSCs, Kupffer cells, LSECs and Liver Bio Kits were purchased directly from Emulate, Inc. (Boston, MA). The cells were kept frozen in liquid $N_2$ storage (vapor phase) until they were cultured. All reagents were aliquoted at appropriate volumes prior to use to avoid multiple freeze-thaw cycles. Fibronectin (Thermo Fisher: Waltham, MA) was resuspended in cell culture grade water to a concentration of 1 mg/mL. Matrigel (Corning; Corning, NY) was thawed overnight on slushy ice in a 4 ˚C refrigerator. Using cold pipette tips, Matrigel was aliquoted to 5 mg aliquots based on the specific stock concentration. L-ascorbic acid (Sigma-Aldrich; St. Louis, MO) was resuspended in cell culture grade water to a concentration of 50 mg/mL. Dexamethasone (Sigma-Aldrich) was resuspended in cell culture grade DMSO (Thermo Fisher) at concentrations of 1 mM and 10 mM. All reagent aliquots were stored at -20 ˚C. The VX used in this study was synthesized and purified by U.S. Army DEVCOM Chemical Biological Center chemists in accordance with international guidelines. VX stock solutions were prepared by dissolving compound in cell culture grade DMSO (Sigma-Aldrich) at a concentration of 5 mM.

### Preparation of liver chips

**LSEC culture.** LSECs were removed from liquid $N_2$ storage and rapidly thawed by placing the cryovial in a 37 ˚C water bath. The contents of the vial were transferred to a 15 mL conical tube containing 3 mL of warm Complete LSEC Culture Medium. The vial was rinsed once with 1 mL of medium, and the 15 mL conical tube was brought up to a volume of 15 mL of Complete LSEC Culture Medium. The cell suspension was centrifuged at 200 x g for 5 minutes at room temperature. The supernatant was then aspirated and the cells were resuspended in 15 mL of fresh Complete LSEC Culture Medium. The LSEC suspension was added to a T-75 flask

and incubated at 37 ˚C and 5% $CO_2$. LSECs were cultured at least 2 days before chip seeding, and medium was exchanged every 2 days.

**Chip activation.** The chips were activated according to the manufacturer's recommended protocol (Emulate, Inc.). Briefly, ER-1 and ER-2 solutions were combined according to Emulate's specified protocol and introduced to the top and bottom channels of the S1-chip. The excess solution was aspirated from the top of the chips and then the chips were activated under constant UV light for 20 minutes. After UV activation, the ER-1 solution was aspirated from both channels and both channels were washed with 200 μL ER-2 solution. The top and bottom channels of each chip were washed with 200 μL of 1X DPBS (Corning) and cold 1X DBPS was left in the channel. The chips were then incubated at 4˚ C overnight.

**Extracellular Matrix (ECM) coating.** The ECM was prepared on ice by combining collagen I (Thermo Fisher) and fibronectin (Thermo Fisher) in ice cold 1X DPBS at a concentration of 100 μg/mL and 25 μg/mL, respectively. After ECM preparation, the cold 1X DPBS was removed from the top and bottom channels of the chips. ECM solution was added to each channel until small droplets formed on the outlets. Droplets of ECM was placed on all 4 ports of each chip, and the chips were incubated at 4 ˚C overnight.

**Hepatocyte seeding.** The chips were equilibrated to room temperature and washed 3X with 200 μL of Complete Hepatocyte Seeding Media. The final wash was left in the top and bottom channels of the chips. Human hepatocytes were removed from liquid nitrogen storage and the cryovial was quick thawed in a 37 ˚C water bath. The thawed cell solution was quickly added to 3 mL of warm Complete Hepatocyte Seeding Medium in a 50 mL conical tube. The vial was rinsed with 1 mL of medium, which was then transferred to the 50 mL conical tube. Complete Hepatocyte Seeding Medium was slowly added to the 50 mL conical tube until the volume was brought to 35 mL. Then, 15 mL of 90% Percoll solution (Sigma Aldrich) in 1X DPBS was slowly layered on top of the 35 mL of cell solution. The cells were centrifuged at 96 x g for 6 minutes at room temperature. The supernatant was carefully aspirated leaving 3 to 5 mL and the pellet undisturbed. The cell pellet was gently resuspended by tilting and rotating the 50 mL conical tube. Complete Hepatocyte Seeding Medium was added to the 50 mL conical tube until the total volume was 50 mL, and the cell solution was centrifuged at 72 x g for 4 minutes at room temperature. The supernatant was carefully removed until the volume was down to 1–2 mL; the pellet was then carefully resuspended as described above. The human hepatocytes were counted in trypan blue solution using a hemocytometer and resuspended to a final cell density of 3.5 x $10^6$ cells/mL in Complete Hepatocyte Seeding Medium. The hepatocyte suspension (50 μL) was then carefully added to the top channel of each chip. After adding the cells, the chips were incubated at 37 ˚C with 5% $CO_2$ for 3 hours (or until the hepatocytes have fully attached). Once the hepatocytes have successfully attached to the channel, a gravity wash was performed on each chip by gently dropping 200 μL of Complete Hepatocyte Seeding Medium on top of both inlet ports of the top and bottom channels. The chips were then incubated overnight at 37 ˚C with 5% $CO_2$.

**Hepatocyte matrigel overlay.** Matrigel was slowly thawed on slushy ice, and preparation of the Hepatocyte Overlay Medium was also performed on ice. Hepatocyte Overlay medium was prepared by diluting the Matrigel into ice cold Complete Hepatocyte Maintenance Medium with chilled tips to a concentration of 250 μg/mL. Chips were removed from the incubator and washed with 200 μL of Complete Hepatocyte Maintenance Medium. 200 uL of Complete Hepatocyte Overlay Medium was pipetted into the top channel of each tip, leaving droplets on both the inlets and the outlets. The chips were then incubated overnight at 37 ˚C and 5% $CO_2$.

**NPC seeding.** The LSECs were harvested from the T-75 using 3 mL of trypsin-EDTA (Sigma Aldrich). 9 mL of warm NPC Seeding Medium was added to the cell suspension and

the entire 12 mL was centrifuged in a 15 mL conical at 200 x g for 5 minutes. The supernatant was aspirated, leaving 100 μL in the tube with which to resuspend the LSECs. The LSECs were counted in trypan blue solution using a hemocytometer and resuspended to a cell density of 9 x $10^6$ cells/mL in cold NPC Seeding Medium. The cell suspension was kept on ice while the other cell types were prepared for seeding. HSCs were removed from liquid $N_2$ storage and the cryovial was rapidly thawed in a 37 ˚C water bath. The thawed cell solution was quickly added to 3 mL of warm NPC Seeding Medium in a 15 mL conical tube. The vial was rinsed with 1 mL of medium, which was then transferred to the 15 mL conical tube. The cell suspension was increased to 15 mL with NPC Seeding Medium, and centrifuged at 250 x g for 5 minutes. The supernatant was aspirated, leaving 100 μL in the tube with which to resuspend the HSCs. The cells were counted in trypan blue solution using a hemocytometer, resuspended to a cell density of 0.3 x $10^6$ cells/mL in cold NPC Seeding Medium, and placed on ice. Kupffer cells were removed from liquid $N_2$ storage and the cryovial was rapidly thawed in a 37 ˚C water bath. The thawed cell solution was quickly added to 3 mL of warm NPC Seeding Medium in a 15 mL conical tube. The vial was rinsed with 1 mL of medium, which was then transferred to the 15 mL conical tube. The cell suspension was increased to 15 mL with NPC Seeding Medium, and centrifuged at 250 x g for 5 minutes. The supernatant was aspirated, leaving 100 uL in the tube with whichh to resuspend the Kupffer cells. The cells were counted as described above and resuspended to a cell density of 1.5 x $10^6$ cells/mL in cold NPC Seeding Medium. The 3 NPC cell suspensions were mixed in a 1:1:1 ratio (v/v/v) in a 15 mL conical tube on ice. 20 μL of the combined NPC suspension was added to the bottom channel of each chip. After seeding, the chips were inverted and incubated at 37 ˚C and 5% $CO_2$ for 4 hours (or until the cells attached). Once the cells were attached to the bottom channel, a gravity wash was performed with 200 μL of Hepatocyte Maintenance Medium for the top channel, and NPC Seeding Medium for the bottom channel. The chips were incubated overnight at 37 ˚C and 5% $CO_2$.

**Chips to Pods and Pods to Zoë.** Complete Hepatocyte Maintenance Medium and NPC Maintenance Medium was warmed in a 37 ˚C water bath for 1 hour. The Medium used for the Pods was gas equilibrated in 50 mL steriflip conical tubes for 5 minutes before use to reduce the risk of bubbles. 300 μL of Complete Hepatocyte Maintenance Medium was added directly over the via (or media inlet port) of the top channel outlet reservoir via and 300 μL of NPC Maintenance Medium was added to the bottom channel outlet reservoir in the same manner. 3 mL of the appropriate medium was added to the inlet reservoirs of each Pod. The Pods were inserted into the Zoë and primed. Once primed, the top and bottom channels of the chips were washed with 200 μL of the appropriate medium and were manually connected to each Pod. The Pods with chips were then placed back into the Zoë and a regulate cycle was run. This is done to stabilize the fluidics of the system. Once the cycle was complete, the flow conditions changed to 30 μL/hour for both top and bottom channels. The top and bottom inlet reservoir medium was replenished every 2 days while the chips were under flow.

## VX solution preparation, dosing, and sampling

The VX used in this study was synthesized and purified (99.1 ± 0.x wt. %) by U.S. Army DEVCOM CBC chemist in accordance with international regulations. Neat VX was then diluted into a working stock of 1 mg/mL in DMSO. Experimental VX dilutions were then prepared from the working stocks by diluting agent in NPC Maintenance Medium without FBS to final concentrations of 6.25 μM, 12.5 μM, and 25 μM. A vehicle control was also prepared by diluting cell culture grade DMSO in NPC Maintenance Medium to a final concentration of 0.6%. The bottom channel inlet reservoir medium was aspirated and replaced with 3 mL of medium containing either VX or vehicle control (DMSO), so that the experimental compound would

be introduced on the endothelial channel (similar to a potential *in vivo* exposure). The Pods were returned to the Zoë and the VX exposure was initiated by increasing the flow rate of the Zoë to 600 μL/h for 5 minutes. The flow rate was then returned to 30 μL/h. After 24 hours, the cells from 12 of the chips were washed 3X with 1X DPBS, trypsinized, pelleted, and flash frozen in liquid $N_2$. VX containing medium was removed from the remaining 12 chips and replaced with untreated medium. The chips were allowed to persist under flow for 7 days before removing the tissues for downstream processing and analysis. Cell pellets were stored at -80˚C before protein, metabolite, and RNA extractions were performed (see below).

## LDH cytotoxicity assay

HepG2 cells (ATCC; Manassas, VA) were cultured in EMEM supplemented with 10% FBS in 24 well plates and incubated for 24 h at 37˚C and 5% $CO_2$. Cells were exposed to varying concentrations of VX diluted in culture medium and incubated for 24 h at 37˚C and 5% $CO_2$. The LDH cytotoxicity assay was performed on the exposed culture medium in accordance with the protocol set out in Thermo Fishers CyQUANT™ LDH Cytotoxicity Assay. All endpoint ODs were read on a FlexStation III multimode plate reader (Molecular Devices, LLC.; San Jose, CA) at wavelengths of 490 nm and 680 nm.

## Human albumin ELISA

The Albumin ELISA was performed in accordance with the protocol set out in Human Albumin ELISA Assay Kit (Abcam). All endpoint optical densities (ODs) were read on a FlexStation III multimode plate reader at a wavelength of 450 nm.

## F-Actin and mitochondrial stain

The NPCs were fixed with 4% paraformaldehyde (PFA) in 1X DPBS for 15 minutes at room temperature. Cells were then permeabilized with 1% saponin (Sigma Aldrich) in 1X DPBS for 30 minutes at room temperature. Blocking was performed by adding 1% BSA and 10% goat serum in 1X DPBS overnight at 4 ˚C. Staining solutions were made by diluting Phalloidin-iFluor 488 (Abcam) or MitoTraker (Thermo Fisher) at 1:500 in blocking buffer. These solutions were then added to the bottom channel of the chips and incubating overnight at 4 ˚C. Hoechst counterstain (Thermo Fisher) was added to the chips at a 1:10,000 dilution for 15 min. The chips were washed 3 times with 1X DPBS and images were obtained using a Keyence BZ-X Fluorescence Microscope (Keyence Corporation of America; Itasca, IL).

## Resuspension and division of cell pellets

Collected cellular pellets were resuspended in 100 μL MS-grade water and pipet-mixed until fully reconstituted. A 50 μL aliquot of each suspension was removed to prep for proteomics and the remaining 50 μL were dried in a speed-vac followed by storage at -80˚C for metabolomics prep and analysis.

## Metabolomics preparation and data analysis

Samples were reconstituted in 410 μL of extraction solution containing isotopically labelled internal standard (ISTD) mixture. The extraction solution was composed of 400 μL of a precipitation solution (8:1:1 acetonitrile: methanol: acetone) and 10 μL of the ISTD mixture (working stock). The ISTD mixture was prepared by making working stocks of each solution at 2 mg/mL by dissolving 10 mg of each standard in 5 mL of 90:10 water: acetonitrile. A working stock solution was prepared by combining the following volumes of each ISTD stock into a

single vial containing 4715 µL of Fisher Optima gold label water with 0.1% FA (final volume 5000 µL):d3-creatine (10 µL), d10-leucine (10 µL), d3-L-tryptophan (10 µL), 13C6-citric acid (20 µL), 13C11-tryptophan (100 µL), 13C6-leucine (10 µL), 13C6-L-phenylalanine (10 µL), T-BOC-L-tertleucine (10 µL), and T-BOC-L-aspartic acid (5 µL). Upon addition of the extraction solution, each sample was vortexed and stored at 4˚C for 60 min to complete protein precipitation. Each sample was centrifuged at 20,000 x g for 10 min at 4˚C to pellet precipitate. 375 µL of the supernatant was transferred to a new tube taking care not to disturb the protein pellet. Each sample was dried to completeness and stored at -80˚C until LC-MS analysis. Immediately prior to LC-MS analysis each fraction was reconstituted in 50 µL of Fisher Optima gold label H2O with 0.1% FA and vortexed briefly. Samples were placed in the refrigerator for 10–15 min to allow for resuspension. Finally, each sample was centrifuged at 20,000 x g for 10 min and transferred into glass autosampler vials (Agilent; Santa Clara, CA) for analysis. Each sample was analyzed on a Thermo Fisher Orbitrap Q Exactive Plus mass spectrometer coupled to a Thermo Fisher Vanquish analytical system. For reverse phase analysis, injections of each sample (5 µL) were resolved at a flow rate of 350 µL/min on a 150 mm x 2.1 mm ID CORTECS T3 2.7 µm column (Waters) with a Phenyl 2.1 mm ID guard column (Phenomenex; Torrance, CA) using a 19 min flow gradient ranging from 100% buffer A (water with 0.1% formic acid) to 95% buffer B (acetonitrile with 0.1% formic acid). For HILIC analysis, injections of each sample (5 µL) were resolved with a flow rate of 350 µL/min on a 100 mm x 2.1 mm ID SeQuant ZIC-HILIC 3.5 µm column (Millipore) with a Phenyl 2.1 mm ID guard column (Phenomenex) using a 15 minute flow gradient ranging from 1% buffer A (90/10 water/acetonitrile with 5 mM ammonium formate) to 37% buffer B (10/90 water/acetonitrile with 5 mM ammonium formate). For both RP and HILIC, data dependent scans were acquired with a resolution of 17,500 with a scan range of m/z 70–1000. AGC target was set to 1E5 with a maximum injection time of 50 ms. All metabolomics data were acquired in positive and negative ionization mode using the heated electrospray ionization source (HESI). The source settings were as follows: spray voltage: 3.50 kV, capillary temperature: 270C, sheath gas (N2): 53 arbitrary units (AU), auxiliary gas (N2): 14 AU, and the probe heater: 350C.

Untargeted metabolomics data analysis was performed with Thermo Scientific Compound Discoverer (ver.3.2.0.421). MS1 features were searched utilizing the Chemspider and Metabolika search nodes at a 5 ppm mass tolerance. MS2 scans were searched utilizing the mzcloud database (searched on January 25th 2021). Detailed search parameters can be found in the supplemental. Results for each chromatographic acquisition were filtered by requiring each feature having a full match in at least 2 sources, MS2 requiring DDA for the preferred ion, Annotation Name not being blank, and then delta ppm between -5 and 5 ppm. The mass spectrometry metabolomics data have been deposited to the Metabolights repository (https://www.ebi.ac.uk/metabolights/) with the dataset identifier MTBLS5466.

## Proteomics prep and analysis

350 µL lysis buffer (4.57% SDS + 57.14 mM ammonium bicarbonate) was added to each sample for a final concentration of 4% SDS and 50 mM ammonium bicarbonate and final volume of 400 µL. Samples were lysed using a Branson probe sonicator at 50% power, with 4 pulses of 5 sec each, with 5 sec of rest between pulses. Samples were then boiled in a heat block at 95˚C for 5 min at 600 rpm, cooled to room temperature, then spun in a microcentrifuge for 10 min, 4˚C, at maximum speed. A Pierce BCA protein assay kit (Thermo Fisher) was used to determine protein concentrations of the clarified lysates.

Protein pellets remaining from the metabolomics crash were combined with the lysate of each corresponding sample to supplement the low protein yield. Probe sonication/boiling was

then repeated as described above. In order to determine a suitable amount of trypsin to use for each sample, protein concentration was roughly estimated to be twice the amount determined in the BCA assay. Samples were then dried in a speed-vac, reconstituted in 200 μL MS-grade water, and ran on an AssayMap Bravo using the in-solution digest application. 80.5 μL denaturant mixture (9.3 M urea + 68.3 mM dithiothreitol) was added to each sample, followed by a 30 min incubation at 56˚C. 10.2 μL of 1 M iodoacetamide was then added to each sample, followed by a 30 min incubation at room temperature in the dark. Trypsin/Lys-C (Promega; Madison, WI) was then added to each sample at a 1:20 enzyme:protein ratio. Samples were then incubated for 60 min at 56˚C.

Digests were desalted on the AssayMap Bravo using the peptide cleanup application and AssayMAP C18 5 μL cartridges (Agilent). Priming and syringe wash buffer (0.1% trifluoroacetic acid in acetonitrile) was used to prime and wash the syringes; equilibration buffer (0.1% trifluoroacetic acid in MS-grade water) was used to equilibrate the C18 cartridges. Samples were eluted in 20 μL elution buffer (0.1% trifluoroacetic acid + 70% acetonitrile + 30% MS-grade water). Elutions were dried in a speed-vac, then resuspended to 0.5 μg/μL in 5% acetonitrile + 0.1% formic acid, vortexed by hand for 2 min, centrifuged at max speed in a tabletop centrifuge for 2 min at 4˚C, and transferred into an autosampler vial for analysis.

Samples were analyzed on a Thermo Eclipse mass spectrometer with FAIMS interface, coupled to an Ultimate3000 HPLC. 6 μL (3 μg total) injections were loaded onto a PepMap 100, 75 μm id × 2 cm C18 trap column (Thermo Fisher Scientific) at 3 μL/min for 10 min with 2% acetonitrile (v/v) and 0.05% formic acid (v/v) followed by separation on an Easy-Spray 75 μm id × 75 cm length, 2 μm, 100 Å C18 column (Thermo Fisher Scientific) at 55˚C. Samples were resolved using a 200 min gradient that began with an increase from 5% to 10% mobile phase B (80% acetonitrile + 0.1% formic acid in water) over the course of 10 minutes, followed by a gradual incline up to 90% mobile phase B at 161 minutes; this was held for an additional 10 minutes before decreasing gradually back down to 5% mobile phase B over the course of 10 minutes. All these gradients were run at a flow rate of 300 nL/min. The Thermo Eclipse was operated in positive polarity and data dependent mode (topN, 3 s cycle time) with a dynamic exclusion of 60 s (with 5 ppm error). FAIMS CVs were set to -50V, -65V, and -85V, with the 3 s cycle time split between each CV. MS1 scan resolution using the Orbitrap was set at 240,000 and the mass range was set to 350 to 2000 m/z. Normalized AGC target was set to 250%, with maximum injection time set to auto. Monoisotopic peak determination was used, specifying peptides and in intensity threshold of 5 x 10^3 was used for precursor selection. Data-dependent MS2 fragmentation was performed using higher-energy collisional dissociation (HCD) at a collision energy of 28% with quadrupole isolation at 1 m/z width. Ion trap scan rate was set to turbo, with a maximum injection time of 35 msec.

Untargeted proteomics data analysis was performed with Thermo Scientific Proteome Discoverer (ver.2.5). Database search was performed with Sequest against the human canonical and isoform fasta, downloaded from Uniprot (March 06, 2020) with an MS1 tolerance of 10 ppm and MS2 tolerance of 0.6 Da [29,30]. A variable protein N-terminal acetylation was also included in the search along with oxidation on methionines and a static carbamidomethylation on cystines. FDR control was achieved utilizing the Percolator node [31]. PTM localization was performed with IMP-ptmRS [32]. Label free quantitation was performed with the Minora feature detection node. Identified proteins from the analysis were then filtered by requiring at least 1 unique peptide, more than 1 PSM and Protein FDR confidence being assigned the level high (0.01 FDR). The mass spectrometry proteomics data have been deposited to the ProteomeXchange Consortium (http://proteomecentral.proteomexchange.org) via the PRIDE database with the dataset identifier PXD037722.

## Transcriptomics sample prep

Cell lysates were prepared on ice, using on-chip lysis using lysis buffer (QIAGEN Buffer RLT with 1% Betamercaptoethanol). Briefly, 175 μL of the lysis buffer was added to each of upper and lower channels and allowed to incubate on ice for 1 min. Lysates were then quickly removed from the two channels, combined in RNAse-free cryotubes, and frozen at -80°C until further processing. RNA was extracted from these samples using the Rneasy® Mini Kit (QIAGEN; Frederick, MD), according to manufacturer's instructions with minor modifications. Briefly, centrifugation was carried out at 10,000 x g and spun for 30s. On-column extraction was performed by incubating 30 μL RNase-free water directly on the spin column membrane for 2 min, followed by elution at 10,000 x g for 1 min. The eluate was run through the column again using the above centrifugation parameters and then stored at -80°C for later analysis. The extracted samples were then quantified on a Qubit™ 4 Fluorometer using the Qubit™ RNA High Sensitivity (HS) Assay Kit following the recommended protocol in the Qubit Assay Kit User Guide. Further quantification and qualification was performed on a 2100 Agilent Bioanalyzer according to the Agilent RNA 6000 Nano Kit Guide. All samples with a RNA Integrity Number (RIN) higher than 8 were used in further analysis.

## RNA sequencing library prep

The Illumina Stranded Total RNA Prep Ligation with Ribo-Zero Plus Kit was used for library generation. The protocol and corresponding calculations were carried out following the Illumina Stranded Total RNA Prep Ligation with Ribo-Zero Plus Reference Guide with minor modifications. Centrifugation steps were at 280 x g for 1 min as opposed to the recommended 10 s due to plate centrifuge limitations. Library samples were then quantified using the Qubit 1X dsDNA High-Sensitivity (HS) Assay Kit and measured on a Qubit™ 4 Fluorometer, using manufacturer's recommended parameters. To assess the quality and quantity during the preparation steps, library samples were measured on a 2100 Agilent Bioanalyzer according to the Agilent High Sensitivity DNA Kit Guide.

## Total RNA sequencing

Each prepared sample library was brought to 4 nM and then 4 μL of each library sample was pooled into one combined "pooled library" sample. Pre-sequencing sample preparation was carried out using the Illumina NextSeq Denature and Dilute Libraries Guide, with minor modifications. Briefly, 2 μL of room temperature 2 N HP3 stock solution was diluted to 0.2 N by the addition of 18 μL of nuclease-free water, 5 μL of pooled library, and 5 μL 4 nM phiX control DNA were placed in separate 1.5 mL tube and diluted HP3 (5 μL) was added to each. The tubes were vortexed and then incubated for 5 min. The following additions below were also made, after a brief vortex and centrifugation. 5 μL of RSB was added followed by 985 μL of pre-chilled HT1 to each tube bringing the denatured libraries and PhiX to 20 pM each. 117 μL of denatured library and PhiX were placed in new 1.5 mL tubes and then 1,183 μL of HT1 was added bringing them to 1.8 pM. 13 μL of the pooled library was removed from the 1.5 mL tube and 13 μL of the denatured PhiX was added. The 1.8 pM final library was placed on ice until sequencing. Library samples were loaded and run on the NextSeq 550 system using the NextSeq 500/550 v2.5 sequencing reagent kit and NextSeq 500 System Guide recommended protocol.

## Transcriptomic data analysis

Following sequencing, demultiplexing, and read generation, the reads were trimmed for adapter sequences using Trimmomatic [33] and filtered for quality using FASTQC [34]. The

reads were then aligned to the human genome using HISAT2 [35]. Next, to assemble transcripts, estimate their abundances, and test for differential expression, we processed the date through CuffLinks [36]. The resulting Fragments Per Kilobase fo transcript per million mapped reads (FPKM) data was analyzed with Qlucore Omics Explorer 3.7 software [37]. The data set was visualized and assessed for differences among groups using multi-group comparisons such as ANOVA. Linear regression analysis was also used to assess the strength of the relationship among exposure levels. Gene Set Enrichments Analysis (GSEA) was utilized through Qlucore to compare the experimentally derived gene set with curated and annotated gene sets from databases such as MsigDB, Reactome, and GO. Further statistical analyses using linear regression were used to define significant gene expression changes within each curated pathway. The RNA sequencing transcriptomics data have been deposited to the NCBI Gene Expression Omnibus (GEO) repository (https://www.ncbi.nlm.nih.gov/geo/), with the dataset identifier GSE221207.

## Supporting information

**S1 Fig. LDH cytotoxicity assay showing the effects of VX on Emulate liver chips after 24 hours.**
(TIF)

**S2 Fig. Boxplot of octyl isocyanate.**
(TIF)

**S1 File. RP positive data set.**
(XLSX)

**S2 File. RP negative data set.**
(XLSX)

**S3 File. HILIC positive data set.**
(XLSX)

**S4 File. HILIC negative data set.**
(XLSX)

**S1 Appendix.**
(DOCX)

## Author Contributions

**Conceptualization:** Jennifer W. Sekowski, Kyle P. Glover.

**Data curation:** Tyler D. P. Goralski, Conor C. Jenkins, Elizabeth S. Dhummakupt, Gabrielle M. Rizzo, Brooke L. Simmons, Alvin T. Liem, Pierce A. Roth, Mark A. Karavis, Jessica M. Hill, Jennifer W. Sekowski.

**Formal analysis:** Tyler D. P. Goralski, Conor C. Jenkins.

**Funding acquisition:** Kyle P. Glover.

**Investigation:** Tyler D. P. Goralski, Daniel J. Angelini, Jennifer R. Horsmon, Elizabeth S. Dhummakupt.

**Methodology:** Tyler D. P. Goralski, Conor C. Jenkins, Daniel J. Angelini, Jennifer R. Horsmon, Elizabeth S. Dhummakupt, Gabrielle M. Rizzo, Jennifer W. Sekowski.

**Supervision:** Tyler D. P. Goralski.

**Validation:** Tyler D. P. Goralski, Conor C. Jenkins, Elizabeth S. Dhummakupt.

**Writing – original draft:** Tyler D. P. Goralski.

**Writing – review & editing:** Tyler D. P. Goralski, Conor C. Jenkins, Daniel J. Angelini, Elizabeth S. Dhummakupt, Jennifer W. Sekowski, Kyle P. Glover.

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
