## [Decision Letter · Decision Letter 0]

27 Jul 2022

PONE-D-22-07224A Novel Approach to Interrogating the Effects of Chemical Warfare Agent Exposure Using Organ-on-a-Chip Technology and Multiomic AnalysisPLOS ONE

Dear Dr. Goralski,

Thank you for submitting your manuscript to PLOS ONE. After careful consideration, we feel that it has merit but does not fully meet PLOS ONE’s publication criteria as it currently stands. Therefore, we invite you to submit a revised version of the manuscript that addresses the points raised during the review process.

Please provide clear responses to each of the reviewer comments.  It is important to be very clear of the research nature of the study given the subject matter and the general public.  

We look forward to receiving your revised manuscript.

Kind regards,

Timothy J Garrett, PhD

Academic Editor

PLOS ONE

Journal Requirements:

2. Please ensure you have included in your Methods section full information regarding the origin (supplier or manufacturer) of all materials used in this study. This includes in particular the origin of the VX chemical warfare agent.

“Funding for this project was provided by the Defense Advanced Research Projects Agency (DARPA) and Defense Threat Reduction Agency (DTRA)-Joint Science and Technology Office (JSTO) for Chemical and Biological Defense.”

“KG

CB10735

Defense Threat Reduction Agency

https://www.dtra.mil/

The funders played a role in the decision to publish.

KG

BQ5066

Defense Advanced Research Projects Agency

https://www.darpa.mil/

The funders played a role in the decision to publish.”

Reviewers' comments:

Reviewer's Responses to Questions

**Comments to the Author**

1. Is the manuscript technically sound, and do the data support the conclusions?

Reviewer #1: Yes

Reviewer #2: Partly

Reviewer #3: Partly

2. Has the statistical analysis been performed appropriately and rigorously? 

Reviewer #1: I Don't Know

Reviewer #2: Yes

Reviewer #3: No

3. Have the authors made all data underlying the findings in their manuscript fully available?

Reviewer #1: Yes

Reviewer #2: Yes

Reviewer #3: No

4. Is the manuscript presented in an intelligible fashion and written in standard English?

Reviewer #1: No

Reviewer #2: Yes

Reviewer #3: Yes

5. Review Comments to the Author

Reviewer #1: I fully admit I am not familiar with the methods used in this research and cannot assess if they were properly executed. The manuscript seems to present a new and ethical way to study the effects of VX on organ systems; it also presents novel effects of VX on liver tissue. However, the use of jargon, combined with problematic sentence structure and poor grammar, made several parts of the results and discussion/conclusions sections difficult to follow. Further, the organization of these sections could be improved. Notably, what the authors consider "method validation" and what they consider new findings are not clearly differentiated.

Reviewer #2: 1. Methods, Hepatocyte Matrigel Overlay (line 432). Authors describe the use of Matrigel for the culturing of their organ-on-a-chip system, but don’t discuss the potential interference in the metabolomics from this highly complex media or if the Matrigel was effectively washed/removed prior to extraction.

2. Discussion (line 316). Authors compare the organ-on-a-chip results as comparable to in vivo results but fail to show sufficient evidence that the low extraction had sufficient metabolic information to derive physiological information relative to the toxicity treatment. More direct comparisons are needed to determine that this system and low extraction are sufficient.

3. Discussion (line 351). Authors again discuss how the results are comparable to the in vivo results but do not show this sufficiently with the interference of the plasticizers.

4. Identifications referenced in Figure 3 are hard to see.

Reviewer #3: This article describes the use of an organ-on-a-chip system to study VX exposure via metabolomics, proteomics, and transcriptomics. The experiments themselves (the proteomics, transcriptomics, and metabolomics) were well-described and experimentally sound. The presentation of the results and the discussion struck me as confusing and incomplete. Statistical analysis was performed, but not presented or discussed in sufficient detail. As a result, the paper needs major revisions prior to publication.

Major Comments:

• Why were the RP negative, HILIC positive, and HILIC negative results not presented or discussed?

• Method details on the partial least squares and the linear regression analysis need to be included. Include an explanation of the color scale in the heat maps

• The metabolomics results description was confusing in that a “top 10” was mentioned, but then only 3 were said to be dysregulated. Please elaborate.

• The phrase “sPLS-DA was performed incorporating time of day” is confusing because time of day is not mentioned previously. Do the authors mean days post-exposure?

• The results/discussion for the plasticizer contamination was unclear. Were they increased in the Day 7 samples as one might expect?

• Regarding days post-exposure, the authors speculated that “One possible answer is that basal level activity within the tissues could mature the longer they are in the chips.” Can’t the untreated controls be used to normalize for such age-associated changes?

• The authors should double check the p-value thresholds. Figure 8, for example, is described as both p=0.005 (caption) and p<0.05 (text)

• My understanding of S-plots was that the extremes were interesting (upper right and lower left). The 10 points highlighted in figure 2 seem to be randomly scattered in the S-plot. Therefore, I do not understand what I am supposed to conclude from the plot and how it relates to the highlighted points.

Minor comments:

• The method section on VX solution preparation cuts off mid sentence

• The caption in figure 4 has B and C backwards

• Page 8: incidents or incidence?

• Page 4: compliments or complements?

• Page 12: “While we observed high p-values”…should this be low p values?

6. PLOS authors have the option to publish the peer review history of their article (what does this mean?). If published, this will include your full peer review and any attached files.

Reviewer #1: No

Reviewer #2: No

Reviewer #3: No

---

## [Author Response · Author response to Decision Letter 0]

12 Sep 2022

PLOS ONE (PONE-D-22-07224)

Response to Reviewers

Authors: Goralski TDP, Jenkins CC, Angelini DJ, Horsmon JR, Dhummakupt ES, Rizzo GM, Simmons BL, Liem AT, Roth PA, Sekowski JW, Glover KP

Title: A Novel Approach to Interrogating the Effects of Chemical Warfare Agent Exposure Using Organ-on-a-Chip Technology and Multiomic Analysis

ID #: PONE-D-22-07224

Comments from the Editor: 

All style requirements have been met. 

2. Please ensure you have included in your Methods section full information regarding the origin (supplier or manufacturer) of all materials used in this study. This includes in particular the origin of the VX chemical warfare agent.

Full information regarding the origin of all materials used in this study, including the supplier and manufacturer, have been included in the manuscript. The origin of the VX chemical warfare agent has been included, as well as the purity of the sample. 

3. Please remove any funding-related text from the manuscript and let us know how you would like to update your Funding Statement.

Funding information was removed from the Acknowledgements section of the manuscript. The information provided in the review is correct. 

4. In your Data Availability statement, you have not specified where the minimal data set underlying the results described in your manuscript can be found. 

All data sets have been uploaded to repositories, and we are awaiting accession numbers. The transcriptomics data were uploaded to Gene Expression Omibus through NCBI, the proteomics data were uploaded to the PRIDE Proteomics Identification Database, and the metabolomics data were uploaded to the Metabolights Database. We will provide you with the accession numbers as soon as possible. 

The data availability statement will remain the same. See above comment. 

6. We note that you have included the phrase “data not shown” in your manuscript. Unfortunately, this does not meet our data sharing requirements.

Figure S1 was added to the manuscript, which shows the LDH cytotoxicity data described.

Reviewer(s)' Comments to Author:

Reviewer: 1

Comments to the Author

1. The use of jargon, combined with problematic sentence structure and poor grammar, made several parts of the results and discussion/conclusions sections difficult to follow. Further, the organization of these sections could be improved. Notably, what the authors consider "method validation" and what they consider new findings are not clearly differentiated.

The manuscript was revised to tone down, or better explain the use of jargon, and correct all problematic sentence structure and poor grammar. Method validation and novel findings were defined in the discussion section. The manuscript states that through validation of this novel methodology, we were able to identify both previously implicated and unique pathways associated with VX exposure. 

Reviewer: 2

Comments to the Author

1. Methods, Hepatocyte Matrigel Overlay (line 432). Authors describe the use of Matrigel for the culturing of their organ-on-a-chip system, but don’t discuss the potential interference in the metabolomics from this highly complex media or if the Matrigel was effectively washed/removed prior to extraction.

The chips were washed 3X with DPBS before the tissues were removed for downstream processing and analysis. All chips, including those that were untreated, received the same lot of Matrigel purchased from Corning (9119020), so any effects on the metabolomics would be equally observed in both control and treated chips. Therefore, these potential effects would not stand out as off target dysregulations in the VX exposed chips. 

2. Discussion (line 316). Authors compare the organ-on-a-chip results as comparable to in vivo results but fail to show sufficient evidence that the low extraction had sufficient metabolic information to derive physiological information relative to the toxicity treatment. More direct comparisons are needed to determine that this system and low extraction are sufficient.

We noted multiple times throughout the manuscript that some of the results obtained from the chip exposures match the dysregulations observed in vivo, with references to previous studies using animal models for VX exposures and downstream omics analysis. We believe that this is validation of our low extraction method. 

3. Discussion (line 351). Authors again discuss how the results are comparable to the in vivo results but do not show this sufficiently with the interference of the plasticizers.

Additional comments were made to the Discussion section to address the potential for plasticizer interference. “Given that the plasticizers are a likely byproduct of the organ chip systems, it is improbable that these off target effects would be observed in multiomics data sets obtained from similar exposures in vivo. However, in our assessment, effects that would implicate the specific plasticizers identified by omics analysis were not reflected in our data.

4. Identifications referenced in Figure 3 are hard to see.

Figure 3 identifications were edited for clarity.

Reviewer: 3

Major Comments to the Author

1. Why were the RP negative, HILIC positive, and HILIC negative results not presented or discussed?

HILIC Positive sPLS-DA data was included in the manuscript. All acquisitions are included as supplemental data files. 

2. Method details on the partial least squares and the linear regression analysis need to be included. Include an explanation of the color scale in the heat maps.

The sPLS-DA methodology is built into the search engine utilized for the metabolomics experiments. Compound Discoverer’s sPLS-DA algorithem is based on the paper entitled Sparse PLS discriminant analysis: biologically relevant feature selection and graphical displays for multiclass problems. This reference has been added to the manuscript. The color mapping of the heatmaps has been declared in the figure legend.

3. The metabolomics results description was confusing in that a “top 10” was mentioned, but then only 3 were said to be dysregulated. Please elaborate.

We have clarified this by stating that out of the ten discriminating compounds, only three were found to be significantly dysregulated.

4. The phrase “sPLS-DA was performed incorporating time of day” is confusing because time of day is not mentioned previously. Do the authors mean days post-exposure?

This was revised in the manuscript. 

5. The results/discussion for the plasticizer contamination was unclear. Were they increased in the Day 7 samples as one might expect?

This was revised in the results section, and the octyl isocyanate boxplot was added to the supplemental. 

6. Regarding days post-exposure, the authors speculated that “One possible answer is that basal level activity within the tissues could mature the longer they are in the chips.” Can’t the untreated controls be used to normalize for such age-associated changes?

7. The authors should double check the p-value thresholds. Figure 8, for example, is described as both p=0.005 (caption) and p<0.05 (text).

The manuscript was revised to reflect that the p-value was 0.005.

8. My understanding of S-plots was that the extremes were interesting (upper right and lower left). The 10 points highlighted in figure 2 seem to be randomly scattered in the S-plot. Therefore, I do not understand what I am supposed to conclude from the plot and how it relates to the highlighted points.

The sPLS-DA algorithem in Compound discoverer places additional factors into its identification of sPLS-DA compounds. It takes into account feature presence across all sample groups and the extremes are some features that appear in few or none of the samples in a sample group thus skewing the discrimination capabilities.

Minor Comments to Author

1. The method section on VX solution preparation cuts off mid sentence • 

The methods section on VX solution preparation was revised.

2. The caption in figure 4 has B and C backwards. 

Figure 4 caption was revised

3. Page 8: incidents or incidence?

Changed to incidence.

4. Page 4: compliments or complements?

Changed to complements.

5. Page 12: “While we observed high p-values”…should this be low p values?

Changed to low p-values.

---

## [Decision Letter · Decision Letter 1]

15 Nov 2022

PONE-D-22-07224R1A Novel Approach to Interrogating the Effects of Chemical Warfare Agent Exposure Using Organ-on-a-Chip Technology and Multiomic AnalysisPLOS ONE

Dear Dr. Goralski,

Thank you for submitting your manuscript to PLOS ONE. After careful consideration, we feel that it has merit but does not fully meet PLOS ONE’s publication criteria as it currently stands. Therefore, we invite you to submit a revised version of the manuscript that addresses the points raised during the review process.

The remaining comments should be readily addressed.  Please respond and provide the necessary edits 

We look forward to receiving your revised manuscript.

Kind regards,

Timothy J Garrett, PhD

Academic Editor

PLOS ONE

Journal Requirements:

Reviewers' comments:

Reviewer's Responses to Questions

**Comments to the Author**

1. If the authors have adequately addressed your comments raised in a previous round of review and you feel that this manuscript is now acceptable for publication, you may indicate that here to bypass the “Comments to the Author” section, enter your conflict of interest statement in the “Confidential to Editor” section, and submit your "Accept" recommendation.

Reviewer #1: (No Response)

Reviewer #2: All comments have been addressed

Reviewer #3: All comments have been addressed

2. Is the manuscript technically sound, and do the data support the conclusions?

Reviewer #1: Partly

Reviewer #2: Yes

Reviewer #3: Yes

3. Has the statistical analysis been performed appropriately and rigorously? 

Reviewer #1: Yes

Reviewer #2: Yes

Reviewer #3: Yes

4. Have the authors made all data underlying the findings in their manuscript fully available?

Reviewer #1: Yes

Reviewer #2: Yes

Reviewer #3: Yes

5. Is the manuscript presented in an intelligible fashion and written in standard English?

Reviewer #1: Yes

Reviewer #2: Yes

Reviewer #3: Yes

6. Review Comments to the Author

Reviewer #1: I agree with the authors that organ-on-a-chip technology has tremendous potential in this space. I still have concerns regarding the discussion of method validation in this paper. It seems the claim that the organ-on-a-chip design is valid and accurate for assessing the effects of VX on liver tissue is based on general similarity of some results with generally established effects of VX. For a proof-of-concept study, I would expect some outlining of explicit parameters, as established a priori, to assess method performance relative to more established cell-line and in vivo methods. While similarity of some results with expected results is good, it is unclear if unexpected results are informative – especially given the high variation observed for the untreated chips and findings regarding carcinogenic plastic contamination. While I do not think the fact that these questions were not answered by the study is in and of itself a problem, I do think the repeated language that the study showed the method is valid and accurate and reveals effects of VX not previously established by other methodologies is problematic. Further, the authors seem to frame this method as a substitute or surrogate for in vivo studies, which is not the intent of organ-on-a-chip technology. It is typically viewed as a bridge between cell lines and in vivo studies.

In sum, the study is promising and innovative, and possible issues are admittedly mentioned, but the authors have used strong language in their conclusions that should be tempered or nuanced at this stage.

Reviewer #2: (No Response)

Reviewer #3: No comments.

This feature does not seem to be working; the system forced me to enter comments and exceed the 100 character limit.:

"If the authors have adequately addressed your comments raised in a previous round of review and you feel that this manuscript is now acceptable for publication, you may indicate that here to bypass the “Comments to the Author” section, enter your conflict of interest statement in the “Confidential to Editor” section, and submit your "Accept" recommendation."

7. PLOS authors have the option to publish the peer review history of their article (what does this mean?). If published, this will include your full peer review and any attached files.

Reviewer #1: No

Reviewer #2: No

Reviewer #3: No

---

## [Author Response · Author response to Decision Letter 1]

19 Dec 2022

Reviewer(s)' Comments to Author:

Reviewer: 1

Comments to the Author

1. I agree with the authors that organ-on-a-chip technology has tremendous potential in this space. I still have concerns regarding the discussion of method validation in this paper. It seems the claim that the organ-on-a-chip design is valid and accurate for assessing the effects of VX on liver tissue is based on general similarity of some results with generally established effects of VX. For a proof-of-concept study, I would expect some outlining of explicit parameters, as established a priori, to assess method performance relative to more established cell-line and in vivo methods. While similarity of some results with expected results is good, it isunclear if unexpected results are informative - especially given the high variation observed for the untreated chips and findings regarding carcinogenic plastic contamination. While I do not think the fact that these questions were not answered by the study is in and of itself a problem, I do think the repeated language that the study showed the method is valid and accurate and reveals effects of VX not previously established by other methodologies is problematic. Further, the authors seem to frame this method as a substitute or surrogate for in vivo studies, which is not the intent of organ-on-a-chip technology. It is typically viewed as a bridge between cell lines and in vivo studies.

In sum, the study is promising and innovative, and possible issues are admittedly mentioned, but the authors have used strong language in their conclusions that should be tempered or nuanced at this stage.

The manuscript was revised to remove and amend all language that would suggest this study was a validation of the method, as well as language which would indicate the method was fully accurate, or proof of concept. 

Reviewer: 2

No Comments to the Author

Reviewer: 3

No Comments to the Author

---

## [Editor Report · Decision Letter 2]

11 Jan 2023

A Novel Approach to Interrogating the Effects of Chemical Warfare Agent Exposure Using Organ-on-a-Chip Technology and Multiomic Analysis

PONE-D-22-07224R2

Dear Dr. Goralski,

We’re pleased to inform you that your manuscript has been judged scientifically suitable for publication and will be formally accepted for publication once it meets all outstanding technical requirements.

Kind regards,

Timothy J Garrett, PhD

Academic Editor

PLOS ONE
---

## [Editor Report · Acceptance letter]

3 Feb 2023

PONE-D-22-07224R2 

A Novel Approach to Interrogating the Effects of Chemical Warfare Agent Exposure Using Organ-on-a-Chip Technology and Multiomic Analysis 

Dear Dr. Goralski:

I'm pleased to inform you that your manuscript has been deemed suitable for publication in PLOS ONE. Congratulations! Your manuscript is now with our production department. 

Kind regards, 

on behalf of

Dr. Timothy J Garrett 

Academic Editor

PLOS ONE